# Gaming in Pandemic Times: An International Survey Assessing the Effects of COVID-19 Lockdowns on Young Video Gamers’ Health

**DOI:** 10.3390/ijerph20196855

**Published:** 2023-09-28

**Authors:** Joanne DiFrancisco-Donoghue, Bernat De las Heras, Orville Li, Jake Middleton, Min-Kyung Jung

**Affiliations:** 1Department of Osteopathic Medicine, New York Institute of Technology, College of Osteopathic Medicine (NYITCOM), Old Westbury, NY 11568, USA; 2School of Physical and Occupational Therapy, Faculty of Medicine, McGill University, Montreal, QC H3A 0G4, Canada; bernat.delasheras@mail.mcgill.ca (B.D.l.H.); orville.li@mail.mcgill.ca (O.L.); 3Indiana Institute of Technology, Fort Wayne, IN 46803, USA

**Keywords:** pandemic, esports, gaming

## Abstract

The onset of COVID-19 coincided with the peak growth of video game usage, with 2.7 billion gamers in 2020. During the pandemic, gaming and streaming platforms offered an entertaining, social, and safe alternative to recreation during severe lockdowns and social isolations. This study aimed to examine the impact of the COVID-19 pandemic on health-related outcomes in self-proclaimed video gamers based on the type of lockdown experienced and to discuss the potential role of video games during times of preventive lockdown measures. This was a cross-sectional international survey constructed by two academic institutions, NYIT (NY, USA) and McGill University (Montreal, Canada), and Adamas Esports (BC, Canada). The survey consisted of questions including demographics, multiple choice, ratings, and Likert scales relating to the periods prior to and during the COVID-19 lockdowns. There were 897 respondents from North America (72.7%), Europe (10.9%), Asia (4.9%), and other countries (11.5%), with a mean age of 22 years. Significant increases in game time were reported in casual and competitive gamers during the first months of the pandemic. The level of gaming, type of lockdown, and physical activity level prior to the pandemic were examined as potential moderating factors. Significant increases in sedentary behaviors (video game time and sitting time) were observed, while physical activity levels remained unchanged in most participants, regardless of the type of lockdown. Sleep time, but not sleep quality, increased, while mental health exhibited opposing effects, influenced by the type of lockdown and gaming competition levels. Video games, when played moderately, could offer a cost-effective, safe strategy to promote socialization and mental health and improve the overall well-being of the non-gaming and gaming population during pandemic times when strict lockdowns are in place.

## 1. Introduction

In 2020, a global pandemic affected territories and regions worldwide, leading to a significant increase in the number of cases and deaths. National and regional governments implemented protective measures to limit the virus’s person-to-person transmission, such as social distancing and restricting the free movement of populations [1]. Due to these COVID-19 restrictions, most economic, social, and recreational activities came to a halt, accelerating society’s transition from in-person to online interactions. This shift transformed many aspects of social and cultural human behavior [2].

This unprecedented situation provided alternative forms of rapidly growing entertainment, like online video gaming, which emerged as a social outlet for many affected by the COVID-19 restrictions. Esports, organized competitions involving multiplayer video games, also grew in popularity. Unlike most traditional sports leagues that had to discontinue or reformat their activities, the online nature of esports facilitated a rapid transition to an entirely virtual competitive environment [3].

Competitive activities cover a broad range, from local cash-prize tournaments to school teams and professional leagues. These activities encompass traditional sports, esports, and other disciplines, offering opportunities for participants, organizers, and fans alike.

The onset of COVID-19 coincided with the peak growth in video game usage, reaching 2.7 billion gamers in 2020. This number is expected to increase to 3 billion by 2023 [4]. The COVID-19 pandemic appears to be one of the main drivers of this upward trend. Gaming and streaming platforms have become increasingly popular as they offer an entertaining, social, and, importantly, safe alternative during severe lockdowns and social isolations [5]. Global reports estimated that total internet traffic increased by 40 to 60% during the spring of 2020, which marked the first wave of the pandemic [6]. Verizon, an American telecommunications company, reported up to a 75% increase in video game usage during the initial weeks of lockdown in the United States. Twitch, the world’s most popular streaming platform for gamers, saw a one-third increase in viewership in March 2020 alone [7].

The uncertain psychosocial context created by the pandemic, coupled with stay-at-home measures, has adversely affected various aspects of well-being, including physical activity levels, mental health, sleep, and social interactions [8,9,10]. Although these restrictions were designed to reduce the spread of the virus, they have also led to several detrimental collateral health consequences among the population [8,9,10].

Remarkably, organizations like the World Health Organization (WHO) have endorsed video games as a means to promote socialization and stress reduction while maintaining social distancing, through campaigns such as #PlayApartTogether [11]. Despite growing evidence of severe adverse health consequences associated with extended lockdown periods, no study has yet explored the effects on video gamers specifically. This study aims to examine the impact of the COVID-19 pandemic on health-related outcomes, including physical activity, gaming time, sleep, diet, and stress, among self-proclaimed video gamers. The authors hypothesized that gaming time and stress levels would have increased, while there would have been a decrease in physical activity and sleep quality, accompanied by worsening dietary practices, among all levels of video gamers.

## 2. Methods

This was a cross-sectional international survey constructed by two academic institutions (NYIT, Old Westbury, NY, USA; McGill University, Montreal, QC, Canada) and a professional esport organization (Adamas Esports Training and Performance, British Columbia, Canada). The New York Institute of Technology approved this study and waived the requirement for informed consent, BHS-1564. The survey was anonymous, with no personal identifiers available. The study data were collected and managed using REDCap (Research Electronic Data Capture) electronic data capture tools hosted at NYIT [12]. REDCap is a secure, web-based application designed to support data capture for research purposes, providing: (1) an intuitive interface for validated data entry; (2) audit trails for tracking data manipulation and export procedures; (3) automated export procedures for seamless data downloads to common statistical packages; and (4) procedures for importing data from external sources.

The Esport COVID-19 electronic survey was created by a multidisciplinary team comprising scientists, academics, esport industry experts, and statisticians. Our research team was responsible for designing, translating, and testing the survey. It was made available in four languages: English, French, Spanish, and Korean. According to the instructions provided at the beginning of the survey, only self-reported video gamers were eligible to participate, regardless of their level of engagement in gaming. The survey featured a multiple-choice question where respondents could identify their level of involvement in esports: Recreational (Esports is a leisure activity, not financially compensated), Amateur (Esports is a leisure activity with occasional small financial compensation), High School (I compete in Esports in a high school setting), College (I compete in Esports in a college setting), or Professional (Esports is my full-time occupation). Before its official distribution, each language version of the survey was pilot tested on a small sample group and further reviewed to ensure accurate translation.

### 2.1. Recruitment and Distribution

The link to the electronic survey was distributed worldwide between 10 June 2020 and 11 October 2020 by various social media outlets, such as LinkedIn™, Twitter™, Facebook™, Discord™, and Slack™, invitation emails, and the general public, who also helped disseminate the research. Participants were able to choose the language of their choice, and ethics and survey instructions were given prior to questions. For this questionnaire, face and content validity were established in all language versions.

The survey consisted of 32 questions using multiple-choice questions, rating questions, and Likert-scale questions (see the Appendix A). For self-proclaimed gamers, the survey inquired about: (a) status of COVID-19 infection, (b) type of lockdown, (c) duration of lockdown, (d) preferred video game genre, (e) gaming competition level, (f) duration and frequency of gaming, (g) duration and frequency of moderate/vigorous physical activity, (h) sitting time, (i) sleep duration and satisfaction, and (j) perceived stress levels. All questions were presented in a differential format, to be answered directly in sequence regarding “prior” and “during” the pandemic.

The participants were divided into casual and competitive gamers. Casual gamers comprised recreational gamers who play video games as a leisure activity with no or small occasional financial compensation, and competitive gamers comprised high school, college, and professional gamers competing in structured esport training programs (see the Appendix A). Further, we divided the levels of lockdown by total, moderate, and light/no lockdown based on the different policies applied by different countries and regions. Total lockdown constituted mandatory stay-at-home measures, leaving home only for necessities, and outdoor physical activities not being permitted. Moderate lockdown constituted mandatory stay-at-home measures, leaving home only for necessities, but with outdoor physical activities being allowed while respecting social distancing. Light or no lockdown was identified as no mandatory stay-at-home measures other than social distancing or no preventive measures at all.

Patient and public involvement—no patients involved.

### 2.2. Data Analysis

All statistical analysis was performed using IBM SPSS Statistics Version 26. A power analysis was performed to estimate a required sample size, for one major aim of this study was to determine whether the confinement increased the gamer’s risk of reducing physical activity. The targeted effect size of OR = 1.5 would indicate that the gamers who changed to be physically active due to lockdown were 50% more than those who changed to be physically inactive due to lockdown. A total of 818 subjects were required to yield 95% statistical power with α = 0.05 in detecting the effect size of OR = 1.5, assuming that their physical activity status was changed due to lockdown in either direction in about 40% of the total sample. The data of 897 respondents were directly exported from REDCap and analyzed for the changes due to lockdown in their gaming time, self-rated PA, sitting time, sleep, and stress and anxiety. The variables were then clustered into subgroups by the types of lockdowns, casual vs. competitive gamers, and active vs. inactive individuals to assess the differences across these subgroups. This study was limited to group comparisons between all factors before and during COVID-19. Descriptive statistics were computed to describe the respondents’ demographic distribution and their responses to the survey questions. Proper statistical hypothesis tests were selected and run to compare the outcomes between groups. For the outcomes of the gamers’ gaming time, PA status, sitting time, sleep hours, sleep satisfaction, and perceived level of stress and anxiety, a McNemar test was used to test the overall changes from before to during the pandemic. The chi-square test was used to test the change between groups: casual vs. competitive gamers, light vs. moderate vs. total lockdown, and active vs. inactive lifestyle. Odds ratios were also computed to estimate the magnitude of changes. Statistical significance was evaluated at α = 0.05.

## 3. Results

This study focused on 897 responses (Figure 1) Overall, the mean age of respondents was 22 years, and 10.3% of the participants identified as women. There were 897 replies from North America (72.7%), Europe (10.9%), Asia (4.9%), and other countries (11.5%). Table 1 presents the demographics.

### 3.1. Physical Activity

How the pandemic impacted on PA based on the recommendations of WHO was analyzed based on subjects meeting the minimum criteria of the WHO recommendation of 150 min of moderate intensity PA in a week, or 75 min of vigorous activity. During lockdown, 24.7% of people who were physically active prior to confinement became inactive during confinement, while 14% who were inactive prior to confinement became physically active during confinement (*p* < 0.001) (Table 2). Furthermore, 45.2% of active gamers became inactive and 30.8% of inactive gamers became active (*p* < 0.001) (Table 3). Gamers were 1.8 times more likely to reduce PA while confined. Prior to confinement, some inactive gamers became physically active, with casual gamers increasing by 14.4% and competitive gamers by 13.5% (*p* = 0.85) (Table 4).

### 3.2. Gaming Time

Overall, the number of hours spent gaming per day increased among both casual gamers (by 73.5%) and competitive gamers (by 75.8%), although the difference was not statistically significant (*p* = 0.14). During moderate lockdown, the increases were 77.6% and 77.2% in total lockdown. In periods of no or light lockdown, the increase in gaming time was 54.9% (*p* < 0.001). Gaming time increased by 78.9% among gamers who were active and by 69.3% among those who were not (*p* = 0.005).

### 3.3. Sitting Time

Overall, the number of hours spent sitting per day increased by 64.3% (*p* < 0.001). Among casual gamers, the increase was 66.6%, while it was 62% for competitive gamers (*p* = 0.45). During moderate lockdown, sitting time increased by 64.7%, and it rose by 68.8% during total lockdown. In periods of no or light lockdown, the increase was 44.9% (*p* < 0.001). Inactive players saw an increase of 58.6%, whereas active players experienced a 69% increase (*p* = 0.013).

### 3.4. Sleep Hours

Overall, the number of hours spent sleeping per day increased for 51.5% of people and decreased for 14.0% during the lockdown period (*p* < 0.001). Among casual gamers, sleep increased by 49.3%, and for competitive gamers, it increased by 54.1% (*p* = 0.36). During moderate lockdown, there was a 54.6% increase in sleep, while in total lockdown, the increase was 51.3%. In periods of no or light lockdown, sleep increased by 40.0% (*p* = 0.013). Active gamers experienced a 53.5% increase in sleep, whereas inactive players saw a 49.1% increase (*p* = 0.012).

### 3.5. Sleep Satisfaction

Overall, sleep satisfaction improved for 35.4% of people and worsened for 22.1% during the lockdown (*p* < 0.001). Among casual gamers, sleep satisfaction improved by 31.6%, while for competitive gamers, it improved for 39.7% (*p* = 0.042). During moderate lockdown, sleep satisfaction improved for 38.1% of people, and in total lockdown, it improved for 34.8%. In periods of no or light lockdown, the improvement was 26.7% (*p* = 0.016). Sleep satisfaction improved for 35.6% of active gamers and for 35.2% of inactive gamers (*p* = 0.24).

### 3.6. Perceived Levels of Stress

Overall, stress levels improved for 40.4% of all gamers but worsened for 46.1% during the lockdown (*p* = 0.32). Among casual gamers, stress improved for 35.9%, while it improved for 45.2% of competitive gamers (*p* = 0.011). During moderate lockdown, perceived stress improved for 44.3% of gamers, and in total lockdown, the improvement was 37.6%. In periods of no or light lockdown, stress improved for 35.0% (*p* < 0.001). The improvement was similar among both active and inactive players, at 40.8% and 39.8%, respectively (*p* = 0.96).

## 4. Discussion

This study examines the impact of the COVID-19 pandemic on self-reported health outcomes among video gamers. We identified significant changes in various health-related outcomes due to the pandemic. Notably, the amount of time spent gaming increased substantially for both casual and competitive gamers during the initial months of the pandemic. We also explored potential moderating factors, including the level of gaming competition, the type of lockdown implemented, and physical activity (PA) levels prior to the pandemic. Lastly, we review and discuss the possible role of video games during lockdowns and their influence on public health during this and future pandemics. Figure 2 represents the authors recommendations during pandemic times based on our results.

### 4.1. The Impact of COVID-19 on Gaming

In this study, 74.6% of all respondents reported an increase in their gaming time during the initial months of the pandemic. These findings align with reports of a 75 to 130% surge in video game engagement during the first weeks of lockdown [7]. Our data also suggest that the amount of time spent gaming was influenced by the severity of the lockdown and by individuals’ levels of physical activity (PA) prior to the pandemic. Interestingly, the level of gaming competition did not appear to have a significant impact.

As might be expected, gamers under moderate and total lockdowns experienced greater increases in gaming time compared to those under light or no lockdown conditions. Surprisingly, those with high pre-pandemic levels of PA also reported greater increases in gaming time than those with lower PA levels. One theory is that physically active individuals viewed video games as a viable alternative form of entertainment when traditional forms of physical activity became less accessible.

These results underscore the role of video games as a source of enjoyment and distraction that aligns with pandemic-related public health guidelines, especially when most traditional recreational activities were unavailable.

### 4.2. The Impact of COVID-19 on Physical Activity and Sitting Time among Video Gamers

The results of this study reveal varying effects of COVID-19 on changes in physical activity (PA) behaviors among self-reported video gamers. Interestingly, 61% of respondents reported no changes in their PA levels during the pandemic, while only 24% indicated reductions in PA. This suggests that the restrictions imposed during the pandemic may have had a minimal impact on the PA levels of gamers. We also examined the influence of pre-pandemic PA levels on changes in PA among video gamers during the pandemic. Based on pre-pandemic PA levels and in accordance with recommended PA guidelines, a majority of gamers were active (54.6%) compared to those who were inactive (45.4%). Active gamers reported a larger decrease in PA (45.2%), whereas inactive gamers reported increases in PA (30.8%).

Interestingly, no significant differences in PA levels were observed between casual and competitive gamers. These findings should be interpreted cautiously, as changes in PA during the pandemic have been shown to be influenced by multiple context-driven factors, including the number of COVID-19 cases per country, the level of lockdown, living environment, age, gender, and the type of PA engaged in [13,14]. For instance, a longitudinal study conducted in China during the initial months of the pandemic revealed a significant decline in time spent on PA (80.6%) [15]. Conversely, a study in Germany reported increases in non-organized PA activities during the strictest lockdown periods [14]. In our study, we found that stricter lockdown measures were associated with larger decreases in PA.

Despite the variations in physical activity (PA) behaviors reported across multiple studies, there is a consensus about the increase in sedentary behaviors during the pandemic [16]. In this current study, most respondents (64.3%) indicated an increase in sitting time. Much like PA levels, the amount of time spent sitting was influenced by the severity of lockdown conditions. Specifically, more respondents reported increases in sitting time during moderate and total lockdowns compared to those in light or no lockdown situations. Notably, gamers who were physically active prior to the pandemic and who experienced moderate to total lockdowns reported the most significant increases in sitting time. This suggests that stricter lockdown measures may particularly exacerbate sedentary behaviors among those who were previously active [14].

### 4.3. The Impact of COVID-19 on Sleep among Video Gamers

Poor sleep is a prevalent issue in modern society, with estimated annual economic costs reaching up to USD 411 billion in the U.S. alone [17]. It has been linked to a wide range of adverse health and social outcomes, including mental health problems [18], risk of cognitive decline and dementia [19], cardiovascular diseases [20], and increased mortality rates [8,9,10,21]. The lockdown measures implemented due to COVID-19, coupled with rising stress, anxiety, and depression, have disrupted normal sleep patterns, impacting sleep quality and contributing to symptoms of insomnia [22,23]. A recent meta-analysis reported that nearly 40% of the population experienced sleep issues during the COVID-19 pandemic [24,25]. In our study, 51.5% of video gamers reported an increase in sleep duration, in contrast to 14% who reported a decrease and 34.5% who reported no change. The data indicate that varying levels of lockdown influenced sleep patterns: those in moderate to total lockdowns reported greater increases in sleep duration but showed mixed results in terms of sleep satisfaction.

The level of gaming competition and pre-pandemic PA levels also significantly affected sleep outcomes during lockdown. Competitive video gamers reported greater improvements in sleep satisfaction compared to casual gamers. Conversely, those who were physically active before the pandemic experienced more significant changes in sleep patterns, with active gamers showing greater fluctuations compared to their inactive counterparts. One speculative explanation could be that competitive gamers, who generally have more structured training routines, were better able to maintain regular sleep–wake cycles, thus improving their overall sleep satisfaction. However, these are speculative hypotheses, and further research is needed to identify other factors that could have influenced sleep patterns among video gamers during the pandemic.

Additionally, evidence suggests that prolonged exposure to video games, especially before bedtime, may have negative effects on various sleep outcomes, such as sleep duration, sleep onset latency, and sleep efficiency [26]. To reap the benefits of video gaming while maintaining healthy sleep patterns during lockdowns, it may be advisable to limit or completely avoid screen exposure before going to sleep.

### 4.4. Gaming and Mental Health during the Pandemic

Playing video games has been suggested as an effective way to promote positive mental health outcomes, including fostering prosocial behaviors and reducing feelings of loneliness [27]. However, excessive gaming can also pose risks, including negative effects on mental health [28]. A recent survey of over 1000 non-gamers revealed significant adverse impacts of COVID-19 home confinement on various mental health outcomes; these effects were linked to social and physical inactivity as well as poor sleep quality [29]. In terms of mental health, our study found that casual gamers fared worse compared to competitive gamers.

When played in moderation—according to studies, this ranges from 1 to 3.5 h daily—video games with cooperative and social interactive features have been shown to positively influence psychological well-being, including reducing levels of stress, anxiety, and depression [30,31]. Despite ongoing efforts to establish an optimal time threshold for gaming that positively impacts well-being, this optimal duration remains elusive.

### 4.5. Strengths and Limitations

The findings from this study offer valuable insights into how varying levels of lockdown impact gaming behaviors, physical activity, sitting time, sleep quality, and mental health among self-reported video gamers. These gamers varied in their levels of physical activity and competitiveness. However, the study does have limitations that warrant careful interpretation of the results. Firstly, due to the self-reported and cross-sectional design of the study, the findings may be subject to recall bias. The respondents were asked to remember pre-pandemic health behaviors, which could compromise the accuracy of the data. Secondly, while our study delved into the role of video gaming during lockdowns, it lacked a non-gamer control group for comparative analysis. To mitigate this, we contextualized our results by comparing them to published studies that focused on non-gaming populations during the COVID-19 pandemic. Thirdly, although our survey had international reach, most participants were from North America. To address this geographic bias, we categorized respondents based on the stringency of their lockdown measures, irrespective of their location. This allowed us to examine the impact of different preventive measures on a range of health outcomes. Fourthly, our study did not employ validated scales to measure mental health outcomes. This decision was made to streamline the survey, but it remains a limitation that should be addressed in future research. Lastly, the demographics of our sample do not fully represent the broader gamer population in terms of age and gender. Future studies would benefit from a more balanced demographic distribution, enabling more nuanced post hoc subgroup analyses.

By acknowledging these limitations, we aim to set the stage for more comprehensive future studies that can build upon our initial findings.

## 5. Conclusions

According to this study, the COVID-19 lockdown measures significantly affected various health-related characteristics among self-reported video gamers. Factors such as pre-pandemic physical activity (PA) levels, competitive gaming involvement, and the extent of lockdown measures all influenced how the pandemic impacted gamers’ health. While further research is needed, the study suggests that during a pandemic—especially when stringent lockdowns are enforced—moderate video gaming could potentially improve the general well-being of both gaming and non-gaming populations.

## Figures and Tables

**Figure 1 ijerph-20-06855-f001:**
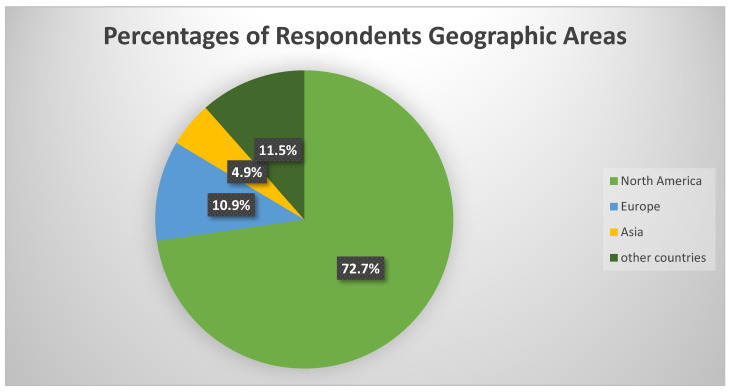
Percentages of respondents according to geographic areas.

**Figure 2 ijerph-20-06855-f002:**
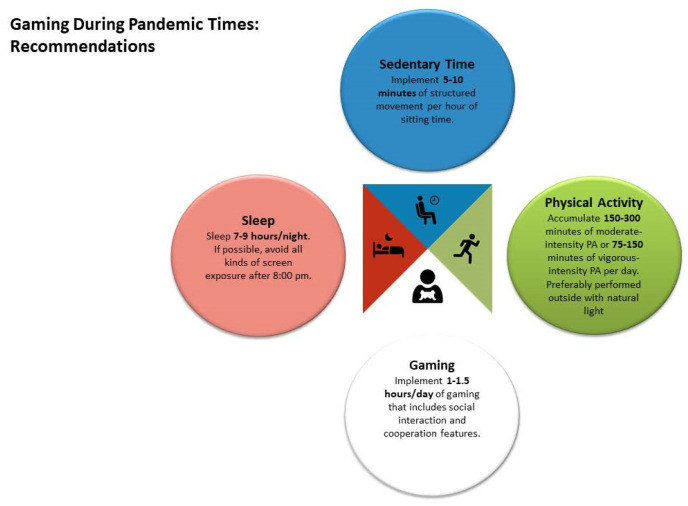
Gaming recommendations during pandemic times.

**Table 1 ijerph-20-06855-t001:** Participant demographics.

Characteristics	Percent
Age	
18	34.6
19–20	21.3
21–25	23.3
26–35	17.2
36–45	3.1
Older than 45	0.4
Sex	
Male	87.2
Female	10.3
Other	2.5
Education	
High school/secondary	45.3
College/undergraduate/CEGEP	39.3
Graduate (master’s/PhD)	11.8
Other	3.6
COVID-19	
I tested positive	1.0
I probably have it	6.6
No, I don’t have it	92.4
Confinement	
Total lockdown	39.9
Moderate lockdown	45.9
Light lockdown	10.3
No lockdown	2.1
Level of Game Competition	
Recreational	52.3
Amateur	18.3
High school	17.2
College	9.9
Professional	2.3
Type of Game Played (Choose all)	
First person (FPS)	57.4
Multiplayer (MOBA)	22.0
Battle Royale	20.3
Massive multi (MMORPG)	17.4
Fighting	9.7
Sports game	8.7
Racing	5.4
Digital Colle (DCCG)	3.3
Continent	
North America	72.7
Europe	10.9
Asia	4.9
Other	11.5

**Table 2 ijerph-20-06855-t002:** Gamers’ lifestyle outcomes before and during quarantine/change from pre- to during quarantine overall.

Gamers’ Lifestyle Outcomes	Before Quarantine	During Quarantine		Change	*p* *
	Percent	Percent		Percent (95% CI)	
Physical activity					<0.001
Physically active	54.6	43.9	Became active	14.0 (11.8, 16.4)	
Not physically active	45.4	56.1	Became inactive	24.7 (21.9, 27.6)	
			No change	61.4 (58.1, 64.5)	
Gaming time					<0.001
0–1 h	6.4	1.6	Increased	74.6 (71.7, 77.4)	
1–3 h	38.3	11.2	Decreased	5.4 (4.1, 7.1)	
3–5 h	32	26.4	No change	20.0 (17.4, 22.7)	
5–7 h	13.7	25.2			
7–9 h	4.9	18.0			
9–11 h	2.1	9.4			
11+ h	2.6	8.3			
Sitting time					<0.001
0–4 h	6.8	2.5	Increased	64.3 (60.7, 67.8)	
4–6 h	13.9	5.5	Decreased	8.5 (6.6, 10.7)	
6–8 h	25.8	15.2	No change	27.2 (24.0, 30.6)	
8–10 h	23.4	20.0			
10–12 h	16.7	22.0			
12–14 h	7.3	18.2			
14–16 h	3.6	10.0			
16+ h	2.4	6.5			
Sleep hours					<0.001
<5 h	5.6	5.1	Increased	51.5 (48.2, 54.8)	
5–6 h	17.5	9.7	Decreased	14.0 (11.9, 16.5)	
6–7 h	30.3	15.5	No change	34.4 (31.3, 37.6)	
7–8 h	34.7	29.9			
8–9 h	9.9	27.9			
>9 h	2.0	11.9			
Sleep satisfaction					<0.001
Very satisfied	15.3	27.2	Improved	35.4 (32.3, 38.6)	

*p* * is the *p*-value as the result of running a chi-square test comparing the proportions between the groups of Before quarantine vs. During quarantine.

**Table 3 ijerph-20-06855-t003:** Change of lifestyle outcomes from pre- to during quarantine compared between Active vs. Inactive.

Lifestyle Outcome	Active	Inactive	*p* *
	Count (%)	Count (%)	
Gaming time			0.005
Increased	377 (78.9%)	276 (69.3%)	
Decreased	20 (4.2%)	27 (6.8%)	
No change	81 (16.9%)	95 (23.9%)	
Sitting time			0.013
Increased	271 (69.0%)	180 (58.6%)	
Decreased	32 (8.1%)	28 (9.1%)	
No change	90 (22.9%)	99 (32.2%)	
Sleep hours			0.012
Increased	257 (53.5%)	195 (49.1%)	
Decreased	77 (16.0%)	46 (11.6%)	
No change	146 (30.4%)	156 (39.3%)	
Sleep satisfaction			0.24
Improved	171 (35.6%)	139 (35.2%)	
Worsened	115 (24.0%)	78 (19.7%)	
No change	194 (40.4%)	178 (45.1%)	
Nutrition satisfaction			<0.001
Improved	76 (15.9%)	83 (20.9%)	
Worsened	167 (34.9%)	86 (21.7%)	
No change	236 (49.3%)	228 (57.4%)	
Stress			0.96
Improved	184 (40.8%)	149 (39.8%)	
Worsened	206 (45.7%)	174 (46.5%)	
No change	61 (13.5%)	51 (13.6%)	

* *p* is the *p*-value as the result of running a chi-square test comparing the proportions between the groups of Active vs. Inactive.

**Table 4 ijerph-20-06855-t004:** Change of lifestyle outcomes from pre- to during quarantine compared between Casual vs. Competitive.

Lifestyle Outcome	Casual	Competitive	*p* *
	Percent (95% CI)	Percent (95% CI)	
Physically active			0.85
Became active	14.4 (11.4, 17.8)	13.5 (10.5, 17.1)	
Became inactive	25.1 (21.3, 29.2)	24.2 (20.2, 28.4)	
No change	60.5 (55.9, 64.9)	62.3 (57.6, 66.9)	
Gaming time			0.14
Increased	73.5 (69.4, 77.3)	75.8 (71.6, 79.8)	
Decreased	4.5 (2.9, 6.7)	6.5 (4.4, 9.2)	
No change	22.0 (18.4, 25.9)	17.6 (14.2, 21.5)	
Sitting time			0.45
Increased	66.6 (61.5, 71.3)	62.0 (56.9, 67.0)	
Decreased	7.9 (5.5, 11.1)	9.1 (6.4, 12.4)	
No change	25.5 (21.2, 30.2)	28.9 (24.4, 33.8)	
Sleep hours			0.36
Increased	49.3 (44.7, 53.8)	54.1 (49.3, 58.9)	
Decreased	14.7 (11.7, 18.1)	13.3 (10.3, 16.8)	
No change	36.0 (31.8, 40.5)	32.6 (28.2, 37.2)	
Sleep satisfaction			0.042 *
Improved	31.6 (27.5, 35.9)	39.7 (35.1, 44.5)	
Worsened	23.7 (20.0, 27.7)	20.3 (16.7, 24.4)	
No change	44.7 (40.2, 49.2)	40.0 (35.3, 44.7)	
Nutrition satisfaction			0.79
Improved	18.6 (15.2, 22.3)	17.9 (14.4, 21.8)	
Worsened	27.9 (24.0, 32.1)	30.0 (25.8, 34.6)	
No change	53.5 (49.0, 58.0)	52.1 (47.2, 56.8)	
Stress			0.011 *
Improved	35.9 (31.5, 40.5)	45.2 (40.3, 50.1)	
Worsened	50.9 (46.2, 55.6)	40.9 (36.1, 45.8)	
No change	13.1 (10.2, 16.6)	13.9 (10.7, 17.6)	

* *p* is the *p*-value as a result of running a chi-square test comparing the proportions between the groups of Casual vs. Competitive.

## Data Availability

The datasets used during the current study are available from the corresponding author on reasonable request.

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
