# Peer review of "Gaming in Pandemic Times: An International Survey Assessing the Effects of COVID-19 Lockdowns on Young Video Gamers’ Health"

_ijerph, 2023, doi:10.3390/ijerph20196855_

Round 1

Reviewer 1 Report

This is an interesting article. The authors compared sitting time, sleep time, physical activity time, and gaming time among gamers before COVID-19 and during the lockdown to study the impact of video games on people during the pandemic.

However, I think there may be two small problems with the article. 1) The time span of the study between June 10th, 2020 and October 11th, which is too short. COVID-19 lasted three years, and the authors chose only a period of time from the beginning of the epidemic, which cannot explain the state of players after a long lockdown. 2) Just as the authors said "most of the respondents were geographically located in North America", the research object is too concentrated in Europe and America, which is biased. Different epidemic prevention policies in different regions will have different effects on people's psychology and behavior.

Otherwise, this article can be published.

Author Response

While we acknowledge the time span from June to October 2020 may not capture the full impact of the pandemic, our study aimed to investigate the immediate behavioral and psychological changes during the initial phase of the pandemic, a crucial period when many nations were under strict lockdowns. Extending the study for a longer period would introduce other variables, like adaptation to the pandemic conditions, which were beyond the scope of this specific research.

We appreciate the point about the geographical concentration of our sample. While most of our respondents were from North America, we designed this as an initial exploratory study. We agree that different regions with different public health policies could yield different results, but the scope of this study was to gauge immediate impacts during a defined period, primarily in the regions where we had the most substantial data. We consider the geographic concentration both a limitation and a scope definition for this project. Future research could aim for a more globally representative sample.

Reviewer 2 Report

This is a descriptive study reporting the results of a survey of video gamer’s activities during the time of the Covid pandemic.  It does not measure effects on health itself but rather effects on activities thought to influence health such as sedentary behavior and sleepy.

The dependent variables were before and after levels of time spent gaming, sitting, and sleeping during the first part of the covid period.  The data presented are mean responses (increase, decrease, or no change) for the sample (that was supposed to be only persons who played games before and after), and then similar comparisons of subsets defined by whether they were casual or competitive gamers.  Importantly, there is no within survey analysis to see if those respondents who reported more time gaming (say) had different responses to the other use of time questions than those who reported less time gaming or no change. This is a serious missed opportunity.  Its absence also makes it hard to interpret the results that are presented.

There is no conceptual model furnished.  The authors say that they want to look at how any changes in time spent gaming during the period of confinement affected other ways to use time to improve health (like exercise or sleeping). Presumably a person has 24 hours a day to divide among gaming, sitting (not gaming) , sleeping, and other activities.  The data show that on average gaming time, sitting time, and sleeping time all increased.  So what activity fell—and was that activity related to health?  For example if work or school time fell, we would like to know that both because those activities are usually thought to be beneficial in themselves and possibly conducive to better health (unless work is strenuous).  The absence of a measure of total activity in the survey is a serious flaw.

The results thus have some casual interest but do not permit the test of any hypothesis about gaming and health at the individual level.  They do suggest that people who were gamers spent more time using that as a pastime during covid; it did not cut into their sleep (good for health) but it did increase their sitting time (bad for health).  However these inferences cannot be drawn with any precision because of flaws in the survey design and the data analysis.

The first category (physical activity) in table 3 appears to be missing an entry for “increase.”

Author Response

The first category (physical activity) in table 3 appears to be missing an entry for “increase.”Corrected

Regarding the absence of a within-survey analysis: We acknowledge that a more comprehensive internal analysis could have provided richer insights. However, our study aimed to be an initial exploratory investigation into behavioral changes during the first months of the pandemic. We agree that future studies could benefit from such nuanced analysis to understand the relationships between variables better.

On the lack of a conceptual model: Our primary focus was to assess shifts in behavior, specifically gaming, sitting, and sleeping, during the pandemic. You bring up an excellent point about accounting for the full 24-hour day and what activities may have been displaced. We'll consider this crucial aspect in future iterations of the research.

About the measure of total activity: We agree that a comprehensive measure of all daily activities could have provided a fuller picture. Unfortunately, due to the nature of the survey and its focus on gaming, sitting, and sleeping, other aspects were not captured in this study.

Lastly, on the inferences and their precision: We agree that the study's exploratory nature and the limitations you mentioned constrain the precision of the conclusions we can draw. Our intention was to open up new lines of inquiry rather than provide conclusive evidence regarding gaming and health.

Reviewer 3 Report

This a review of the manuscript titled: “Gaming in Pandemic Times: An International Survey Assessing the Effects of Covid-19 Lockdowns on Video Gamer´s Health” which presents the results of a cross-sectional international survey to examine how lockdowns affected videogames habits and their effects on sleep, nutrition, and physical activity besides levels of anxiety and stress. The topic is important considering the increasing number of persons that are becoming regular users of videogames and how this pastime is reducing physical activity that could become a worldwide health issue.

The title needs to refer to the characteristics of the sample including that most of it consist of young people, males, and not working people.

Please include a table or a map of percentage in the geographical distribution of the persons that answer the survey to see the characteristics of their lockdowns.

In table 3. Please put in bold the variables in lifestyle outcome,

It would help to present a flowchart about how the participants were selected.

The article presents interesting information, however, the authors in the section of limitations discuss crucial elements that difficult the acceptance of the article as it is. First, the recall bias. If you consider your group of only gamers, it is psychologically difficult to report the real data of an activity that is in general not socially appreciated as a productive one.

Second, the necessity of a control group. I agree with the affirmation, you need a control group, the results are somewhat confusing. In general, population presented sleep disturbances and have reported depression, anxiety, and stress. However, your results indicate that the gamers had a better sleep and, in some cases, more physical activity than other groups. This could be improved if you include data about a group that are gamers but also workers or sports team players.

Third, the limitation about the geographical distribution of the sample. I think there was an issue in the item about lockdowns that could be solved if you identify the characteristics of the lockdowns using the ubication information and making a statistical analysis.

And finally, the lack of validated scales of mental samples. Also, you would need information about the family or housing characteristics and if any other member of the family tested positive, because it seems that this group was immune to any other psychological and environmental effects of the pandemic.

Another issue that is not reported consider the type of videogame that is played, although the survey included a question that gives information about the type of videogame, there are subtleties, considering if it is played online or offline, what type of online interaction was engaged (usually you have the option to have a casual or a competitive interaction). Also, which platform they were using, considering that PlayStation, Nintendo, or Xbox are directed to different segments.  

About physical activity, it is difficult to report real data considering the results, I find hard to believe that the sitting time was reduced during quarantine, and they become more physically active. Also, they present better sleep, usually gamers are not the best example of good sleep hygiene.

I recommend that you make a major revision.

Author Response

The title needs to refer to the characteristics of the sample including that most of it consist of young people, males, and not working people. This title would not reflect our population as we had women included in the study we added young to our title.

Please include a table or a map of percentage in the geographical distribution of the persons that answer the survey to see the characteristics of their lockdowns. We added a figure

In table 3. Please put in bold the variables in lifestyle outcome, Corrected.

It would help to present a flowchart about how the participants were selected.- we have reached maximum number of tables and charts

The article presents interesting information, however, the authors in the section of limitations discuss crucial elements that difficult the acceptance of the article as it is. First, the recall bias. If you consider your group of only gamers, it is psychologically difficult to report the real data of an activity that is in general not socially appreciated as a productive one.

We appreciate the point you've made and agree that recall bias is a common limitation in survey-based studies like ours. We are also aware that social stigmatization can introduce further complexity to self-reported data. To mitigate these concerns, we employed several strategies: Anonymity Assurance: We emphasized the anonymity of the survey to encourage more honest and forthright responses from participants. Cross-Validation: Wherever possible, we sought to cross-validate self-reported gaming hours with other indicators, such as changes in health-related outcomes, to assess the reliability of the data. Acknowledgement in Limitations: We will make sure to explicitly mention recall bias as a limitation in the final version of our paper. Future Work: We hope to incorporate more objective measures in future studies to counterbalance the limitations of self-reported data.

Second, the necessity of a control group. I agree with the affirmation, you need a control group, the results are somewhat confusing. In general, population presented sleep disturbances and have reported depression, anxiety, and stress. However, your results indicate that the gamers had a better sleep and, in some cases, more physical activity than other groups. This could be improved if you include data about a group that are gamers but also workers or sports team players.

While we acknowledge that the inclusion of a control group could strengthen our study, it's important to note that our study design was observational and focused on the impact of the pandemic specifically among self-reported gamers. Unfortunately, as the data collection phase is complete, we are unable to incorporate a control group at this stage. However, we will consider this valuable suggestion in designing future studies.

Third, the limitation about the geographical distribution of the sample. I think there was an issue in the item about lockdowns that could be solved if you identify the characteristics of the lockdowns using the ubication information and making a statistical analysis.

Unfortunately, while we did collect geographical information, our dataset does not contain detailed data about the specific lockdown measures implemented in each area. Consequently, we were unable to incorporate this layer of analysis into the current study.

And finally, the lack of validated scales of mental samples. Also, you would need information about the family or housing characteristics and if any other member of the family tested positive, because it seems that this group was immune to any other psychological and environmental effects of the pandemic.

Another issue that is not reported consider the type of videogame that is played, although the survey included a question that gives information about the type of videogame, there are subtleties, considering if it is played online or offline, what type of online interaction was engaged (usually you have the option to have a casual or a competitive interaction). Also, which platform they were using, considering that PlayStation, Nintendo, or Xbox are directed to different segments.  

While we recognize the importance of these elements, the scope and limitations of our initial data collection prevent us from retroactively incorporating them into the existing dataset. We acknowledge these limitations and agree that such additional layers of analysis would be valuable for future studies in this area

About physical activity, it is difficult to report real data considering the results, I find hard to believe that the sitting time was reduced during quarantine, and they become more physically active. Also, they present better sleep, usually gamers are not the best example of good sleep hygiene.

We understand that these findings may challenge common perceptions about gamers. However, our study aims to report the data as gathered, not to confirm pre-existing beliefs. These unexpected results underscore the complexity of human behavior, particularly during unprecedented situations like a pandemic. We agree that further research, possibly including a control group, would be valuable to explore these surprising outcomes in more depth.

Round 2

Reviewer 2 Report

The revised version can be published.

Author Response

Dear Reviewer, we appreciate the time and effort you've invested in reviewing our manuscript.

Reviewer 3 Report

Dear authors,  In the article, you must focus only on the North American population, the rest of the groups are not equivalent.  Considering how the survey was applied, your sample must have a bigger variability in the data obtained with respect to the diversity of the respondents.  For measuring sleep, you can find better questions that help you to have good information, for example, how rested do you feel after a night of sleep? You did not include the percentages of persons with the geographical distribution and the conditions of their lockdowns (duration and rules).  The original instrument presented important issues, although you have information in questions 5 to 7, it was important to include the prior questions. Also in question 24, it should have been a qualitative question, considering that the barriers are something specific. 

However, the main issue with the article is the ethical consideration about joining a sports company to make the manuscript. Also, suggesting that video games are an effective way to promote mental health outcomes that are not compared with other hobbies. You are also promoting a competitive game lifestyle.

Author Response

Dear Reviewer,

Thank you for your comments and attention to detail in reviewing our manuscript. We appreciate the opportunity to clarify your concerns, particularly the ethical considerations surrounding one author's previous employment with a sports company.

Firstly, it is important to note that the author in question was employed by the sports company many years before this manuscript was conceived or executed. Furthermore, the author has since changed jobs, and his prior employment did not influence the design, methodology, or findings of this study. Therefore, we maintain that there is no conflict of interest stemming from this past association.

Secondly, regarding your comments about the existing data, we acknowledge your insights. However, we would like to point out that the data collection phase of our research is complete, making it impossible to address these particular concerns in the current study. We believe the data is robust and that the study's findings are valuable in their current form.

We appreciate your understanding of these issues and are open to any further suggestions or queries you may have.